# A Drastic Shift in Lipid Adducts in Colon Cancer Detected by MALDI-IMS Exposes Alterations in Specific K^+^ Channels

**DOI:** 10.3390/cancers13061350

**Published:** 2021-03-17

**Authors:** Jone Garate, Albert Maimó-Barceló, Joan Bestard-Escalas, Roberto Fernández, Karim Pérez-Romero, Marco A. Martínez, Mª Antònia Payeras, Daniel H. Lopez, José Andrés Fernández, Gwendolyn Barceló-Coblijn

**Affiliations:** 1Department of Physical Chemistry, University of the Basque Country (UPV/EHU), 48940 Leioa, Spain; jone.garate@gmail.com (J.G.); R.fernandez@imgpharma.com (R.F.); josea.fernandez@ehu.es (J.A.F.); 2Institut d’Investigació Sanitària Illes Balears (IdISBa, Health Research Institute of the Balearic Islands), 07120 Palma, Spain; albert.maimo@ssib.es (A.M.-B.); joanbe88@gmail.com (J.B.-E.); karim.perez@ssib.es (K.P.-R.); marco.martinez@ssib.es (M.A.M.); mariaantonia.payerascapo@ssib.es (M.A.P.); danielhoracio.lopezlopez@ssib.es (D.H.L.); 3Research Unit, Hospital Universitari Son Espases, 07120 Palma, Spain; 4Research Department, IMG Pharma Biotech S.L., BIC Bizkaia (612), 48160 Derio, Spain; 5Pathology Anatomy Unit, Hospital Universitari Son Espases, 07120 Palma, Spain; 6Gastroenterology Unit, Hospital Universitari Son Espases, 07120 Palma, Spain

**Keywords:** colorectal cancer, lipidomics, ion adducts, potassium channels

## Abstract

**Simple Summary:**

Colorectal cancer (CRC) is one of the most preventable yet deadliest cancers, one reason being that it involves very different lesions. Currently, there is a great international effort to improve CRC classification using as many molecular features as possible. A cutting-edge technique, imaging mass spectrometry, is used to enable the visualization of the bidimensional (2D) distribution of molecules across tissues in order to study how the composition of the cell membrane, in particular membrane lipids, changes in tumors. Our previous studies indicate that lipid composition is highly sensitive to cell alterations. Importantly, during the analysis, we are also able to establish changes in charged lipids, observations that can be misinterpreted. A close study of our results alongside information from public databases leads to the identification of gene coding for a potassium channel that could account for our observations and could represent a suitable target for drug development.

**Abstract:**

Even though colorectal cancer (CRC) is one of the most preventable cancers, it is one of the deadliest, and recent data show that the incidence in people <50 years has unexpectedly increased. While new techniques for CRC molecular classification are emerging, no molecular feature is as yet firmly associated with prognosis. Imaging mass spectrometry (IMS) lipidomic analyses have demonstrated the specificity of the lipid fingerprint in differentiating pathological from healthy tissues. During IMS lipidomic analysis, the formation of ionic adducts is common. Of particular interest is the [Na^+^]/[K^+^] adduct ratio, which already functions as a biomarker for homeostatic alterations. Herein, we show a drastic shift of the [Na^+^]/[K^+^] adduct ratio in adenomatous colon mucosa compared to healthy mucosa, suggesting a robust increase in K^+^ levels. Interrogating public databases, a strong association was found between poor diagnosis and voltage-gated potassium channel subunit beta-2 (KCNAB2) overexpression. We found this overexpression in three CRC molecular subtypes defined by the CRC Subtyping Consortium, making KCNAB2 an interesting pharmacological target. Consistently, its pharmacological inhibition resulted in a dramatic halt in commercial CRC cell proliferation. Identification of potential pharmacologic targets using lipid adduct information emphasizes the great potential of IMS lipidomic techniques in the clinical field.

## 1. Introduction

Colorectal cancer (CRC) as a whole is the third most common malignant cancer and is the fourth main cause of cancer death worldwide [1]. Even though CRC is one of the most preventable cancers, it is currently one of the deadliest. Furthermore, although the implementation of screening programs has led to a significant reduction in CRC incidence in the population aged >50 years in high-income countries, the incidence in people <50 years has unexpectedly increased in a significant manner for causes as yet unknown [2]. Thus, it is imperative to improve prevention and treatment strategies since current approaches are failing. The development of new techniques for tumor classification based on their molecular characteristics through genomic analysis and multiparameter cytometry has enabled the selective application of therapeutic strategies adapted to each tumor, helping to improve personalized therapy [3]. Some of these strategies, such as microsatellite instability [4,5], detection of mutated B-RAF or K-Ras [6], are already implemented in many hospitals and have brought about an improvement in some cancer type outcomes and the development of targeted therapies based on the specific molecular features of each tumor.

The urgent need for an improved stratification system has led to the formation of international consortiums aiming to establish a new classification of CRC based on their precise molecular biology [7,8,9]. This ambitious challenge is currently being approached from different disciplines involving transcriptomics, proteomics, or epigenetics, while membrane lipidomics remains a rather unexplored strategy. There is no doubt that to develop true predictive biomarkers in solid tumors, it is necessary to apply comprehensive approaches in order to enable the characterization of all cell types present in a tumor, including the stromal compartment [10] and its microenvironment [11]. In this context, imaging mass spectrometry (IMS) techniques offer the possibility of analyzing tissue sections by averting cell-type isolating procedures. For this reason, they are steadily becoming a tool of reference for classifying tissues according to their differential composition at a resolution that currently reaches cellular level [12,13,14].

Among all the biomolecules currently analyzed by IMS, lipids have turned out to be particularly suitable for these techniques. Lipids are a broad, heterogeneous family of biomolecules highly intermingled at the metabolic level but differing substantially at the structural one. These intrinsic features have long impeded the development of analytical techniques that could efficiently cope with lipid diversity [15,16]. However, owing to the development of several MS-based technologies, including IMS, it is possible to obtain extensive lipidomes not only from lipid extract but also from tissue sections. The results obtained demonstrate that the cell lipidome is not only greatly cell-type dependent but also highly sensitive to any pathophysiological alteration such as differentiation or tumorigenesis [17]. In this context, the analysis of healthy and tumor colon mucosa by matrix-assisted laser desorption ionization (MALDI)-IMS at 10 µm of lateral resolution demonstrated that the epithelium has a different lipidome from the stroma and that the impact of pathology on a particular lipid species differs according to cell type [18,19,20]. Furthermore, a detailed analysis of the results unveiled the strict regulation of the lipidome along the colon epithelium, which was concomitant with the differentiation status of the colonocytes [19,20]. Interestingly, these results were obtained by assessing the changes in the total levels of particular lipid species. However, during MALDI-IMS analysis, particularly in positive-ion mode, the formation of ionic adducts of the lipid species is rather common, of which H^+^, K^+^, and Na^+^ are the most frequent [21]. While adduct formation complicates lipid identification in IMS, it may serve as a sensor of changes in the relative concentration of cations as a result of alterations in cell homeostasis. Of particular interest is the [Na^+^]/[K^+^] ratio, which functions as a biomarker for homeostatic alterations in a model of spinal cord injury [22] and tumor cell viability in a xenograft model [23,24]. Based on all this evidence, the impact of colon tissue malignization on the adduct distribution of membrane lipid species detected in positive-ion mode ought to be studied, regardless of the impact the malignization had on total PC species levels.

In this study, we demonstrated a drastic and solid shift of the [Na^+^]/[K^+^] adduct ratio in colon adenomatous polyp mucosa compared to healthy mucosa, strongly suggesting a robust increase in K^+^ tissue levels. We hypothesized that this shift was a consequence of an altered function of the ion channels accounting for K^+^ transport. Due to the large number of channels involved either directly or indirectly in regulating K^+^ levels, we first interrogated public databases on gene expression, finding a strong association between altered gene expression levels and poor prognosis for KCNAB2 (voltage-gated potassium channel subunit beta-2). We further investigated the impact of the inhibition of Na^+^ and K^+^ channels on commercial CRC cell growth. The results showed a dramatic halt in CRC cell proliferation when KCNAB2 was pharmacologically inhibited. Furthermore, KCNAB2 expression levels were solidly associated with a poor prognosis. In fact, this gene turned out to be an interesting pharmacological target in an oncological context as it was found overexpressed in three of the CRC molecular subtypes defined by the consensus molecular subtype (CMS) [25]. The identification of potential pharmacologic targets using lipid adduct information generated by MALDI-imaging techniques strongly emphasizes the great potential of these often underutilized techniques.

## 2. Materials and Methods

### 2.1. Human Colon Biopsy Collection

Ethics statement: The sample collection for this study was specifically approved by the Ethics Research Committee of the Balearic Islands (IB 2118/13 PI (colon biopsies) and IB 4035/19 PI (surgical specimens)). Informed consent was obtained in writing from each patient before performing each endoscopy or surgery.

Human colon biopsies were obtained in the Endoscopic Room of Hospital Universitari Son Espases (Palma, Spain). Endoscopic biopsies were resected using Radial Jaw standard-capacity biopsy forceps (Radial Jaw^TM^ 4, Boston Scientific, Marlborough, MA, USA) and immediately snapped frozen in liquid nitrogen and saved at −80 °C until sample preparation. Sections of ~10 μm thickness were prepared using no cryoprotective substances or embedding material in cryostat (Leica CM3050S, Leica Biosystems, Wetzlar, Germany) at −20 °C and placed on plain glass microscope slides. A consecutive section was stained with hematoxylin and eosin (Sigma-Aldrich, Madrid, Spain) for structure identification.

Isolation of human colon crypts was performed using surgical biopsies. Biopsies were washed with standard phosphate buffer saline and cut into 1–2 cm fragments using scissors. Colon tissue was digested using 10 mM DTT for 5 to 10 min and then 8 mM EDTA (in HBSS without calcium or magnesium (Thermo Fisher Scientific, Waltham, MA, USA)) for 45 min at 4 °C. Crypt detachment was achieved by vigorously shaking the suspension. The supernatant was collected and centrifuged at 100× *g* for 10 min at 4 °C.

### 2.2. Sample Preparation for MALDI-Imaging and Data Analysis

Reagents: 2-mercaptobenzothiazole (MBT) matrix, as well as hematoxylin and eosin for histological staining, were purchased from Sigma-Aldrich. Leucine enkephalin acetate hydrate, ammonium acetate, and sodium hydroxide solution were purchased from Sigma-Aldrich Chemie (Steinheim, Germany).

Detailed information regarding the MALDI-imaging and data analysis can be found in Garate et al. [18]. Briefly, MBT was used as a matrix for positive-ion detection and was deposited with the aid of a glass sublimator (Ace Glass 8023) [18,26]. Four sections of healthy mucosa and three sections of adenomatous polyp mucosa from different individuals were scanned in positive-ion mode, using the orbitrap analyzer of an LTQ-Orbitrap XL (Thermo Fisher), equipped with an N_2_ laser (100 µJ max power, elliptical spot, 60 Hz repetition rate). Mass resolutions of 30,000, 60,000, and 100,000 at *m*/*z* = 400 Da were used to record the data, and the scanning range was 480–1000 Da.

Spectra were analyzed using dedicated software (MSI Analyst, Noray Bioinformatics S.L., Derio, Biscay, Spain). A detailed description of data processing may be found in ref [18] but, briefly, the spectra were normalized using a total ion current (TIC) algorithm and aligned using the Xiong method during the parsing stage [27]. All peaks with intensity values lower than 0.5% of the intensity of the strongest peak were filtered to reduce the number of *m*/*z* and speed up the analysis.

Lipid assignment was based on a comparison between experimental *m*/*z* and the species in the software’s database (<33,000 lipid species plus adducts) and in the LIPID MAPS^®^ database (www.lipidmaps.org (last accessed on 1 February 2021)). Mass accuracy was always better than 9 ppm and was typically better than 3 ppm. In this type of analyzer, the mass accuracy depends somehow on the intensity of the peaks and, thus, higher intensity *m*/*z* present better mass accuracy. If a mass channel contained a contribution from more than one species, the percentage due to each species was determined using the [H^+^]/[Na^+^]/[K^+^] ratio found for each lipid class and assumed to be constant for all the species in a given class [23,28,29]. This ratio was calculated for each individual spectrum with the aid of an in-house built program. The correction cannot be performed just by deconvolution of the average spectrum, as it may change for each type of tissue [23,28,29].

### 2.3. Cell Culture Experiments

#### 2.3.1. Cells and Reagents

HT29 and SW480 CRC commercial cell lines were purchased from the European Collection of Authenticated Cell Cultures (ECACC, Salisbury, UK) and were grown in a humidified cell culture incubator (37 °C, 5% CO_2_). HT29 and SW480 cells were cultured in MEM (Labclinics, Barcelona, Spain, Ref. L-0415-500) and DMEM (Labclinics, Barcelona, Spain, Ref. L-0103-500), respectively. All the culture media were supplemented with 10% FBS (Labclinics, Barcelona, Spain, Ref. S181B-500) and 1% penicillin/streptomycin (Labclinics, Barcelona, Spain, Ref. L-0018-100).

3,4-dihydroxyphenylacetic acid (DOPAC) inhibitor of K^+^ channels (Sigma-Aldrich Chemie, Darmstadt, Germany) was dissolved in dimethyl sulfoxide (DMSO; Sigma-Aldrich Chemie) at a stock concentration of 50 mg/mL and stored in darkness at room temperature.

#### 2.3.2. Pharmacological Inhibition of K^+^ Channels

Prior to inhibiting experiments, gene expression levels of KCNAB2 were established in HT29 and SW480 CRC commercial cell lines (Appendix A). Cell viability for the commercial cell lines HT29 and SW480 was assessed using the Cell Proliferation Kit I (MTT, Roche, Merck, Darmstadt, Germany). Cells were plated in 24-well plates at the density of 9 × 10^4^ cells/well and grown in the presence or absence of DOPAC (500 µM) or 0.01% DMSO (vehicle control) for 24 and 48 h. The inhibitors were freshly diluted in the media prior to the treatment. Following the first 24 h of treatment, 1 mL of media was removed, and 100 µL of MTT labeling reagent was added to each well and cultured for another 3 h. Then the solution was carefully aspirated, and 200 µL of MTT solubilization buffer was added into each well and homogenized. Absorbances were measured at 500 nm using a spectrophotometric microplate reader (Synergy H1 Hybrid Multi-Mode Reader; BioTek Instruments, Inc., Winooski, VT, USA).

#### 2.3.3. Clonogenic Survival Assay

Colony-formation assay: Cells were diluted to 100 and 200 cells/well with supplemented medium and seeded out in six-well plates to test for clonogenic survival. Colonies were grown in a humidified cell culture incubator (37 °C, 5% CO_2_). Cells were incubated in the presence or absence of DOPAC (500 µM) or 0.01% DMSO (vehicle control) for ten days. The inhibitors were diluted in fresh media prior to treatment. All culture media were replaced every two days. At the end of the experiment, cells were fixed and stained in agitation with a 5% glutaraldehyde 0.5% crystal violet buffer at 4 °C for 30 min. Pictures were taken using a Canon EOS M50 camera with a 55 mm to 200 mm lens.

### 2.4. Interrogation of CRC Patient Gene Expression Data Sets

#### 2.4.1. Interrogation of Patient-Derived CRC Biopsies

The following human gene expression datasets are publicly available in the NCBI GEO [30] and The Cancer Genome Atlas (TCGA) datebases: GSE44076, GSE39582, GSE20916, GSE97689, GSE5206, and TCGA colon and rectal cancer (COADREAD) RNAseq dataset, for potassium and sodium channels subunits.

#### 2.4.2. Interrogation of Macro- and Micro-Dissected Datasets

In order to clarify the gene expression profile of sodium and potassium channels in carcinoma cells and healthy control cells, we have interrogated the GSE20916 (*n* = 105 macro-, *n* = 40 micro-dissected) [31] and GSE35602-6480 (*n* = 30 CRC, *n* = 4 control both micro-dissected) [32] gene expression array data. Differential expression analysis was independently calculated in each dataset for the different colorectal cancer-associated group samples versus control samples by using the NCBI-GEO2R [33,34,35].

#### 2.4.3. Interrogation of Fluorescent Activated Cell Sorting CRC Cells Datasets

Some of the obtained results were validated in GSE34053 and GSE39396 datasets. The Transcriptome Analysis Console Software (version 4.0.1, Affymetrix, Santa Clara, CA, USA) was used for the data analysis.

#### 2.4.4. Patient Survival Analysis and CMS Association

Patient overall survival and disease-specific survival Kaplan–Meier plots of the different target genes were performed using UCSC Xena Browser on primary tumor samples of patient cohorts TCGA COAD and COADREAD [36], according to gene expression levels (RNAseq–Illumina HiSeq) over two groups and quartile sample aggrupation.

Overall survival (OS), according to the NIH, is defined as the length of time from either the date of diagnosis or the start of treatment for a disease, such as cancer, that patients diagnosed with the disease are still alive. While the disease-specific survival rate (DSSR) is defined as the percentage of people in a study or treatment group who have not died from a specific disease in a defined period of time, the time period usually begins at the time of diagnosis or at the start of treatment and ends at the time of death. It is important to emphasize that the patients who died from causes other than the disease being studied are not counted in this measurement. Hence, OS and DSSR convey slightly different information as in OS, the cause of death does not have to be necessarily linked to the disease, whereas in DSSR, it is. For the Kaplan–Meier survival analysis, the log-rank test (test statistics and *p*-value) was conducted using two common methodologies: two groups of samples or comparing upper and lower quartile.

Association of target genes and consensus molecular subtype (CMS) labels [25] were calculated and visualized for TCGA COADRED samples using UCSC Xena Browser [36].

### 2.5. Statistics

GraphPad Prism (version 8.0) software was used for statistical analyses comparing adducts levels percentage and in cell experiments. The specific statistical test used for a particular graph is indicated in the figure legend. The overall survival graphs were calculated in Xena browser, which uses a log-rank test to compare Kaplan–Meier curves and reports the test statistics (*χ*^2^) and *p*-value (*χ*^2^ distribution).

## 3. Results

### 3.1. Colon Mucosa Malignization Induces a Drastic Shift in the [Na^+^]/[K^+^] Adduct Ratio

Three types of tissue can easily be distinguished in human colon mucosa: the epithelium, made up of a single-cell monolayer of colonocytes; the lamina propria (or stroma), composed of different cell types such as fibroblasts and immune cells; and the muscularis mucosae (Figure 1). The colon epithelium invaginates into the stroma, generating the functional units called crypts. At the bottom of these structures reside the adult stem cells that divide and differentiate into fully mature colonocytes while ascending the crypt.

In previous studies, we showed how a series of lipid species, particularly phosphatidylinositol (PI) and phosphatidylethanolamine (PE) plasmalogen species, as well as arachidonic-containing species, were gradually distributed along the colon crypt. Colonocytes regulate the levels of these species concomitantly to the differentiation process in a strict and precise manner by a yet-to-be-defined mechanism [19,20]. Most of the phosphatidylcholine (PC) species levels (84% of total PC), however, remain constant regardless of the differentiation state of the colonocyte [19]. In this context, it was previously shown that changes in the [Na^+^]/[K^+^] adduct ratio detected during MALDI-IMS measurements are associated with profound changes in tissue homeostasis [22,23,24]. Taking all these facts into account, we decided to investigate first whether the [Na^+^]/[K^+^] PC adduct ratio changed according to the colonocyte differentiation status.

Hence, the changes in ion adducts occurring along a colon crypt were evaluated in healthy mucosa pixel-by-pixel. Similar to previously [19,20], we depicted a path from the base to the top of the colon crypt and followed the H^+^, Na^+^, and K^+^ adduct levels for each of the eight PC species detected (Figure 2 and Appendix A). Overall, the results showed that none of the adducts displayed a differentiation-dependent signature. Next, we calculated the [Na^+^]/[K^+^] ratio, which was approximately 0.5:0.5 for all the PC species. This value was observed even in the few PC species showing a gradient along the crypt, such as PC 32:0 and PC 38:4 [19]. Then, we evaluated the impact of epithelium malignization on adduct distribution following a similar approach. Consistent with the results in the healthy epithelium, none of the adducts changed their level along the adenomatous epithelium (Figure 3 and Appendix A). However, a robust, consistent increase was observed in K^+^ adduct levels, which led to a shift of the [Na^+^]/[K^+^] ratio from approx. 0.5:0.5 to approx. 0.3:0.7. Importantly, this shift was observed in all PC species regardless of their specific alterations observed in the total content between adenomatous (AD) and healthy epithelium (Figure 4). Thus, while total PC 34:1 levels increased in AD epithelium, total PC 36:2 and PC 36:3 decreased, but in both cases, the [Na^+^]/[K^+^] ratio was 0.3:0.7 [19].

Next, we investigated whether this shift was specific for epithelial cells. A path was depicted across the stromal compartment from the basal to the luminal side of the mucosa, evaluating the changes in both adduct levels and calculating the [Na^+^]/[K^+^] ratio in healthy and adenomatous sections (Figure 4 and Appendix A). The results mimicked the ones observed for the epithelium, i.e., the individual levels of the adducts did not change along the lamina propria, but the [Na^+^]/[K^+^] ratio did shift from 0.5:0.5 in healthy lamina propria to 0.3:0.7 in malignant lamina propria.

In a previous study, we compared the changes between healthy and adenomatous polyps in total PC species [19]. For this purpose, we added the contribution of each of the adducts of a particular PC species, i.e., the total PC 34:1 level was the result of adding PC 34:1 K^+^, PC 34:1 Na^+^, and PC 34:1 H^+^, and so on for each PC species. According to this study, in the epithelium, PC 34:1 showed a robust and significant increase in malignant tissue compared to healthy tissue, while the level of species such as PC 38:4, 38:5, or 36:3 showed a moderated decrease. Finally, the level of some PC species such as PC 36:1 or 36:2 remained unaltered. Taking this into account, as well as the solid shift in potassium adducts for all the species (Figure 3 and Figure 5), we investigated how the K^+^ adducts levels changed proportionally in each of the PC species detected (Figure 4 and Appendix A). The results showed that, regardless of the behavior that the total species showed, the percentage of K^+^ adducts was always increased in the adenomatous compared to the healthy mucosa (Figure 4b, red part).

Finally, we investigated whether this shift in [Na^+^]/[K^+^] ratio was specific for PC species by assessing the changes in this ratio in sphingomyelin (SM) species, the second most abundant membrane lipid family detected in MALDI-IMS in positive-ion mode. In this case, only one SM species was consistently detected in the colon sections analyzed, SMd34:1. The results demonstrate that SM-adducts behaved identically to PC adducts, indicating that the alterations in the [Na^+^]/[K^+^] adduct ratio was not lipid class-dependent (Figure 6).

In summary, all the evidence points to a general increase in K^+^ adducts across the malignant mucosa. Taking into account the fact that the increase in K^+^ adducts correlates with an increase in K^+^ levels [22], these results strongly suggest that tumorigenesis could be tightly associated with a deregulation of the mechanisms tightly controlling K^+^ ion levels.

### 3.2. Impact of Colon Cancer on K^+^ and Na^+^ Channels at the Transcriptional Level

Increasing evidence suggests that ion channels play important roles in cell proliferation, migration, apoptosis, and differentiation [37]. In particular, K^+^ channels play a major role in the maintenance of plasma membrane potential in a hyperpolarized state, essential in the gastrointestinal epithelium that must continuously transport mass quantities of water, electrolytes, and nutrients. K^+^ channels are the largest and most diverse group of ion channels in the human genome, having 77 genes coding them [38]. Taking this diversity into account, several transcriptomic databases were interrogated to investigate the distribution of Na^+^ and K^+^ channels in colon mucosa and then to assess the impact of CRC on the level of these channels. The first approach gave rise to several gene candidates that were able to account for the deregulation of K^+^ concentration levels in malignant tissue. One of the reasons for this multiplicity in genes could be related to sample heterogeneity of CRC biopsies in terms of cell typology.

However, currently, it is possible to delve into this heterogeneity by using available databases analyzing more specific samples, either as micro-dissected epithelium or sorted cells [10,31,32,39]. First, we analyzed two databases: GSE20916 [31] generated using both macro- and micro-dissected biopsies, and GSE35602-6480 [32] generated using only micro-dissected biopsies. By doing so, we retrieved a list of five Na^+^ and K^+^ channel subunits that were differentially expressed on a more consistent basis (Figure 7). In tumor epithelial samples (light and dark blue in Figure 7), potassium voltage-gated channel subfamily A regulatory beta subunit (*KCNAB2*) and potassium calcium-activated channel subfamily M regulatory beta subunit 4 (*KCNMB4*) genes appeared upregulated, while *KCNJ8* was also expressed in the stroma fraction (bright green bar in Figure 7). Analogously, *SCN9A* and *SCNN1B* down-regulation was highly associated with tumor epithelial cell-enriched fractions.

These observations were confirmed in two datasets carried out using only tumor cells isolated by cell sorting, GSE34053 and GSE39396 [10,39]. In particular, the GSE34053 dataset analyzed cancer-associated fibroblasts (CAF, tumor stromal cells) and CD133+ cells (tumor epithelial cells) [39]; while CD133+ cells expressed higher levels of *KCNAB2* and *KCNMB4*, CAF cells (stroma) overexpressed *SCN9A* and *KCNJ8* subunit channels (Appendix A). Finally, we interrogated the GSE39396 database, which includes a wide range of cell types: epithelial (EpCAM+), endothelial (CD31+), leukocytes (CD45+), and CAFs (FAP+) isolated by FACS from tumor biopsies [10]. In this array, *KCNJ8* showed a higher expression in CAFs (FAP+) compared to tumor epithelial (EpCAM+) cells (Appendix A).

Next, we investigated the relationship between the overall survival (OS) and disease-specific survival rate (DSSR) of CRC patients and the expression levels of these particular channel subunits (Figure 8and Appendix A). Using the Kaplan–Meier plots, we could establish that the expression of *KCNAB2*, *KCNJ8*, *KCNMB4*, and *SCN9A* appeared significantly associated with a CRC poor prognosis in at least one of the comparisons made (2 groups or upper and lower quartile) and using OS or DSSR data (OS/2-groups Figure 8a, OS/quartile Figure 8b, DSSR/2-groups Figure 8c, DSSR/quartiles Figure 8d). Importantly for this study, *KCNAB2* was the gene displaying the best association, using both OS and DSSR data and both analytical approaches (Figure 8). Unexpectedly, *SCN9A* high expression, and not lower expression as suggested in Figure 7, was associated with CRC poor prognosis for disease-specific survival data (Figure 8b, lower row). Taking these results into account, *KCNAB2* turned out to be an interesting target to further explore.

We further investigated how these results would fit in the CRC consensus molecular subtype (CMS) classification [25] using the UCSC Xena Browser software [36]. The CMS is a large international consortium aiming to establish a consensus classification based on RNA expression data. Currently, this classification is subdivided into four CMS groups with distinct biological characteristics: CMS1 (MSI immune), CMS2 (canonical), CMS3 (metabolic), and CMS4 (mesenchymal). Figure 9a shows the expression level heat-map of *KCNAB2*, *KCNJ8*, *KCNMB4*, *SCN9A,* and *SCNN1B* genes in healthy (normal) and primary tumors samples, classified according to the CMS labels. First, we confirmed the higher expression of *KCNAB2* in primary tumors compared to a healthy sample using the TCGA COADREAD database (Figure 9b). Next, we used the CMS classification to evaluate the expression of *KCNAB2*, *KCNJ8*, *KCNMB4*, *SCN9A,* and *SCNN1B* in each of the above-mentioned subtypes (Figure 9c–g, respectively). According to this system, CRC tumors falling into the MSI immune and mesenchymal subtypes are the ones with the worst prognosis, while the canonical subtype is the largest in representation, accounting for approximately 40% of CRC samples [25]. Figure 9c shows that *KCNAB2* was overexpressed in tumor samples in these three subtypes, while in the metabolic subtype (CMS3), this gene showed rather similar expression levels in tumor and healthy tissues. Further, *KCNJ8* and *SCN9A* showed a suitable association with the mesenchymal subtype (Figure 9d,f, respectively), while *KCNMB4* showed similar expression levels between subtypes (Figure 9e). Finally, *SCNN1B* expression levels were significantly lower in all subtypes, particularly in CMS1 (Figure 9g).

Altogether, the described overexpression of KCNAB2 in tumor epithelium, its solid and consistent association with a poor CRC prognosis, and its association with the three molecular subtypes (CMS1, 2, and 4), which despite having a very different pathway enrichment account for a large percentage of CRC heterogeneity, makes this gene particularly interesting for pharmacological inhibition.

### 3.3. Pharmacological Inhibition of KCNAB2 Drastically Halts Colon Cancer Proliferation

Based on these in-silico results, we investigated the potential role of KCNAB2 in tumor cell proliferation using two commercial colon cancer cell lines were used, in particular, HT29 and SW480 cells. Before performing these experiments, we established *KCNAB2* gene expression levels in HT29 and SW480 CRC commercial cell lines, which were clearly overexpressed in CRC cells compared to healthy colonocytes (Appendix A).

CRC cells were grown in the presence or absence of a pharmacological inhibitor of KCNAB2, in particular, 3,4-dihydroxyphenylacetic acid (DOPAC, 500 µM) [40]. DOPAC is a metabolite of the dopamine commonly found in mammals and the intestinal microbiota, as a fermentation product of the quercetin [41,42], which was demonstrated to have antioxidant and antiproliferative properties [43]. According to several studies in cells and rats, DOPAC shows no apparent toxicity at the tested doses [43,44].

The MTT assay was used to determine the impact of DOPAC on cell proliferation rate after 24 and 48 h of treatment. The results showed a profound impact of DOPAC treatment on cell growth, which decreased in HT29 cells by 65.0% and 71.1% at 24 and 48 h, respectively, and in SW480 cells by 46.6% and 90.9% (Figure 10a). While this assay is commonly used as an indicator of cell viability, proliferation, and cytotoxicity, the MTT assay measures cellular metabolic activity. For this reason, we ran a clonogenic assay or colony-formation assay, which is based on the ability of a single cell to grow into a colony. The results clearly revealed the profound impact of DOPAC treatment on the growth of both HT29 and SW480 cells (Figure 10b,c). Altogether, these in vitro studies indicate that KCNAB2 could be playing an important role in CRC cell homeostasis, particularly affecting cell proliferation.

Altogether, these results clearly confirm the critical role of KCNAB2 in the proliferation of commercial CRC cells. The latter is in agreement with published evidence showing that ATP-sensitive K^+^ channel activators, such as minoxidil and diazoxide, inhibit the proliferation of several human colon cancer cell lines (SW1116, LoVo, Colo320DM, and LS174t). Conversely, several Kv blockers (dequalinium, amiodarone, and glibenclamide) cause a marked growth-inhibition [45,46,47].

## 4. Discussion

The word cancer involves multiple diseases that may differ considerably in their diagnosis, prognosis, and treatment, even when sharing the same affected organ. Consequently, there is great interest in thoroughly describing tumor tissues, not only at the molecular level but also at the single-cell level, as the heterogeneity in terms of cell composition is more extensive than initially expected. In this scenario, imaging mass spectrometry techniques offer a promising opportunity to describe tumor protein and lipid composition. Using MALDI-IMS to characterize the human colon mucosa lipidome, we demonstrate that it is possible to differentiate, based merely on their specific lipidomes, not only between healthy and tumor biopsies but also between the epithelial and stromal compartments, reaching in some cases a subcellular resolution [18,19,20,48]. Consistently, the usefulness of describing the lipidome by IMS to discriminate between healthy and pathological biopsies has been demonstrated in a wide diversity of tumor types, including ovarian [49], thyroid [50], oral squamous cell carcinoma [51], breast [52], brain [53], and liver [54].

During the analysis of samples by MALDI-IMS in positive-ion mode, most lipid species are detected as ion adducts, the most common of which are potassium and sodium adducts. While changes in ion adduct levels are commonly associated with alterations in lipid metabolism, the results showed herein indicate that this may not always be the case. In fact, the mechanisms underlying the formation and metabolism of these lipid adducts, as well as their biological meaning, remain largely unknown. In this study, we show a drastic shift in the ratio of Na^+^/K^+^ adducts in malignant colon mucosa compared to healthy tissue. The thorough analysis throughout all PC lipid species strongly indicate that this shift was due to a net increase in K^+^-adducts and depended neither on lipid family, as it was observed in both PC and SM species, nor on the particular lipid species, as it affected all PC species (e.g., PC 34:1, PC 32:0…). It is important to emphasize that the increase in K^+^ levels occurred regardless of what the particular PC species underwent during colonocyte differentiation or malignization (i.e., increase, decrease, or remaining constant, Figure 4). This observation implies that the mere increase of an ion adduct does not necessarily have to imply an alteration in lipid metabolism, and it could be associated, for instance, with an ionic redistribution.

In previous studies, it was demonstrated that the increase in [Na^+^]/[K^+^] ratio is associated with a net increase in potassium levels [22,23]. Thus, using MALDI-IMS, Fernández et al. showed an alteration in the [Na^+^]/[K^+^] balance following spinal cord traumatic injury [22]. Importantly, this modification was confirmed by X-ray fluorescence, demonstrating the ability of IMS to determine the [Na^+^]/[K^+^] ratio using the relative intensity of the peaks of Na^+^ and K^+^ adducts of a given lipid species. It is well known that lipids detected by MALDI-IMS are mostly those that have previously migrated to the matrix layer [55]. Therefore, if the Na^+^/K^+^ adduct ratio reflects the tissue [Na^+^]/[K^+^], it would indicate that the lipid migration to the matrix layer is already in the adduct form.

Ion gradients are one of the major biological driving forces underlying life-sustaining processes, and for this reason, cells count on a formidable collection of ion channels and transport proteins, enabling the strictest control and regulation of ion concentration. Ion transport has been studied thoroughly in the context of neurotransmission, muscle, and heart contraction, while its role in cancer is less understood. However, recent studies show that the activities of several ion channels are also important to support both malignant and non-malignant cell processes such as cell proliferation, cell migration, or the phenotypic switch from an epithelial state to a mesenchymal phenotype [56,57,58,59,60]. Changes in expression levels of K+ channels can occur at the genomic, transcriptional, post-translational, or epigenetic level, and in some cases, a quantitative increase in activity can be explained by upstream changes. Our results showing the large increase in K^+^-lipid species adducts observed in the adenomatous mucosa would be consistent with these findings and point toward an altered function of the mechanisms regulating K^+^ transport across cell membranes.

At the clinical level, the evidence that K^+^ channels are key actors in tumorigenesis is steadily piling up, and, consistently, the dysregulated expression of these channels is associated with a poor prognosis in a variety of human cancers [61,62,63,64]. The thorough interrogation conducted herein of human gene expression databases using CRC biopsies pinpointed KCNAB2, a regulatory subunit of a K^+^ channel, as a suitable candidate to be a key player in the malignant transformation of colon tissue. Importantly, overall survival and disease-specific data in CRC patients robustly support this observation (Figure 8). Interestingly, using the CMS classification, we determined that this gene was overexpressed in two of the subtypes with the worst outcome, the mesenchymal and immune subtypes, and in the CMS2 (or canonical) subtype, which accounts for the largest number of CRC patients (ca. 40%). Taking into account all this information, together with the drastic growth halt in CRC cell lines treated with KCNAB2 inhibitor, this gene appears as a very suitable target for drug development.

In this context, different ion transport molecules, including K^+^ channels, are already being studied as targets for cancer treatment [64]. Thus, the expression of Kv10.1, a K^+^ channel encoded by the *KCNH1* gene, is ectopically expressed in over 70% of solid human tumors and seems to participate in the regulation of key processes of tumorigenesis [61]. Importantly, the aberrant expression of Kv10.1 seems to originate from the altered expression of relevant regulators of cancer cell phenotypes, such as p53, E2F1, and miR-34a [61]. According to a recent study, the inhibition of Kv10.1 expression or function leads to mitochondrial fragmentation, an increase in reactive oxygen species, and autophagy. However, K^+^ channels are highly variable and, depending on the specific channel, the inhibition of certain channels might have an opposite effect. KCNQ1, which codes for Kv7.1 K^+^ channels, is frequently downregulated in hepatocellular carcinoma cell lines and tissues, and hepatocellular carcinoma patients with lower KCNQ1 expression have a poor prognosis. Mechanistically, KCNQ1 can interact with β-catenin to affect its subcellular distribution and subsequently reduce the activity of Wnt/β-catenin signaling, which further blocks the expression of its downstream targets [65]. Interrogation of this particular gene in the GSE35602_6480 and GSE20916 databases results in a slight overexpression of colon tumor tissue compared to healthy tissue. As could not have been otherwise, deregulation of ion transport in cancer offers a highly complex scenario to interact in and more research is needed in order to fully understand which channel (or channels) would be better to target.

## 5. Conclusions

According to the present study, the [Na^+^]/[K^+^] ratio emerges as a suitable source of pathophysiological biomarkers. Thus, we have previously established changes in this ratio in injured rat spinal cord [22], tumor xenografts as a response to treatment [23,24], and herein, as a result of tissue malignization. Importantly, although the presence of altered levels of K^+^ adducts in tumor tissue has been consistently demonstrated in a variety of cancers [51,66,67,68,69], these alterations were commonly associated with changes in lipid metabolism, while the ionic contribution is rarely taken into account. Hence, we have demonstrated that, in addition to providing information regarding lipid metabolism, IMS lipidomics may also yield relevant information regarding ion transfer, which we consider might often have been underestimated.

## Figures and Tables

**Figure 1 cancers-13-01350-f001:**
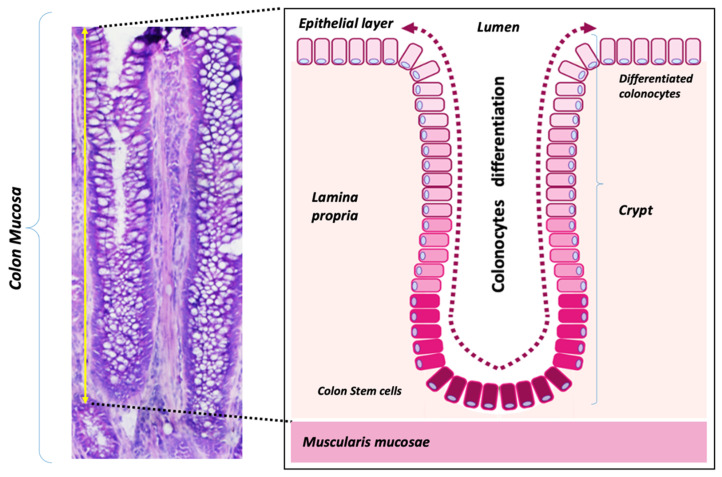
Human colon mucosa tissues. (**left**) Histological sections of a human colon biopsy stained with hematoxylin-eosin showing the three types of basic tissues present in the colon mucosa: the epithelial layer (EP), the lamina propria (LP), and the muscularis mucosae (MM); (**right**) Schematic diagram of a colon crypt, consisting of a single-cell layer of epithelial cells (colonocytes) that invaginates into the lamina propria (stroma). Adult stem cells (in dark red) divide and migrate upward toward the lumen; during this migration, cells undergo a differentiation process to mature colonocytes.

**Figure 2 cancers-13-01350-f002:**
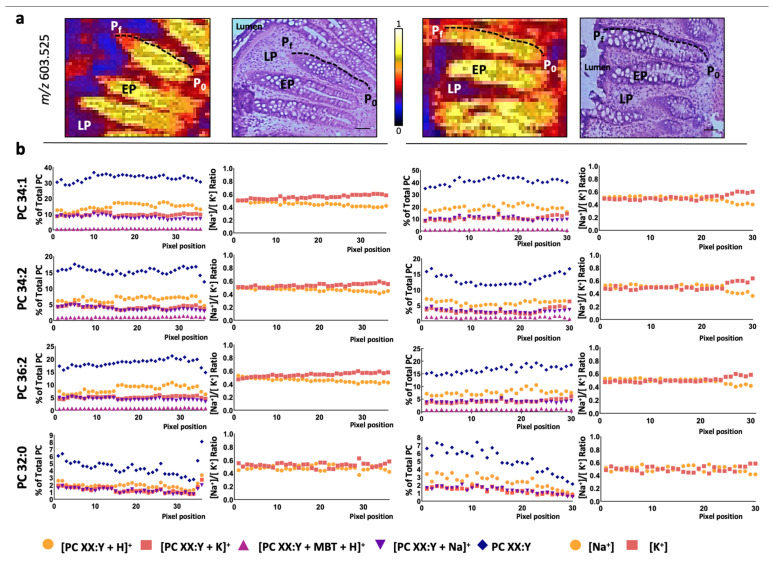
PC adduct distribution in healthy colon epithelium. (**a**) Selected MALDI-IMS images obtained during the analysis in positive-ion mode (in particular, *m*/*z* = 603.525) showing the paths depicted from the base (P_0_) to the top (P_f_) of the colon crypt to analyze, pixel-by-pixel, the changes in the lipidome along the healthy epithelium. Hematoxylin-eosin images of the consecutive sections are included for comparison. Scale bar = 100 μm. (**b**) Individual distribution of PC adducts along the depicted paths and the [Na^+^]/[K^+^] ratio for selected PC species (PC 34:1, PC 34:2, PC 36:2, and PC 32:0). The rest of the species (PC 36:1, PC 36:3, PC 36:4, and PC 38:4) may be found in Appendix A. EP: epithelium; LP: lamina propria; P_0_: first pixel of the path; P_f_: final pixel of the path.

**Figure 3 cancers-13-01350-f003:**
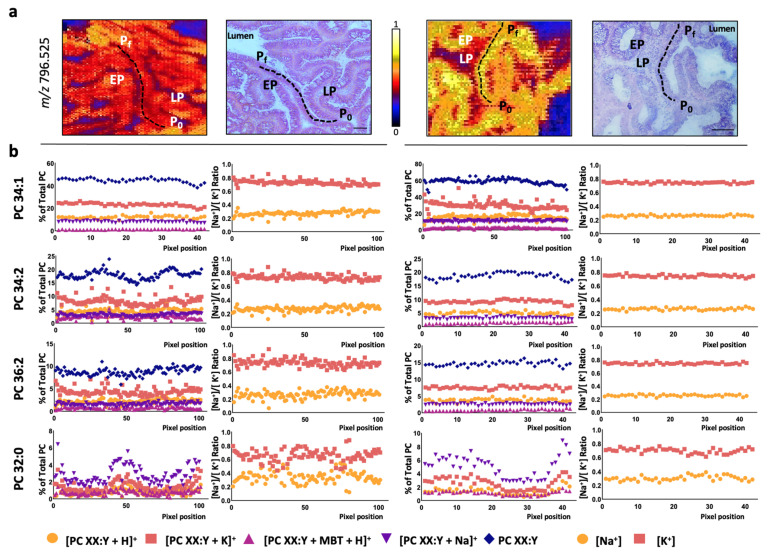
PC adduct distribution in adenomatous colon epithelium. (**a**) Selected MALDI-IMS images obtained during the analysis in positive-ion mode (in particular, *m*/*z* = 796.525) showing the paths depicted from the basal (P_0_) to the luminal side (P_f_) to analyze, pixel-by-pixel, the changes in the lipidome along the adenomatous epithelium. Hematoxylin-eosin images of the consecutive sections are included for comparison. Scale bar = 100 μm. (**b**) Individual distribution of PC adduct along the depicted paths and the [Na^+^]/[K^+^] ratio for selected PC species (PC 34:1, PC 34:2, PC 36:2, and PC 32:0). The rest of the species (PC 36:1, PC 36:3, PC 36:4, and PC 38:4) may be found in Appendix A. EP: epithelium; LP: lamina propria; P_0_: first pixel of the path; P_f_: final pixel of the path.

**Figure 4 cancers-13-01350-f004:**
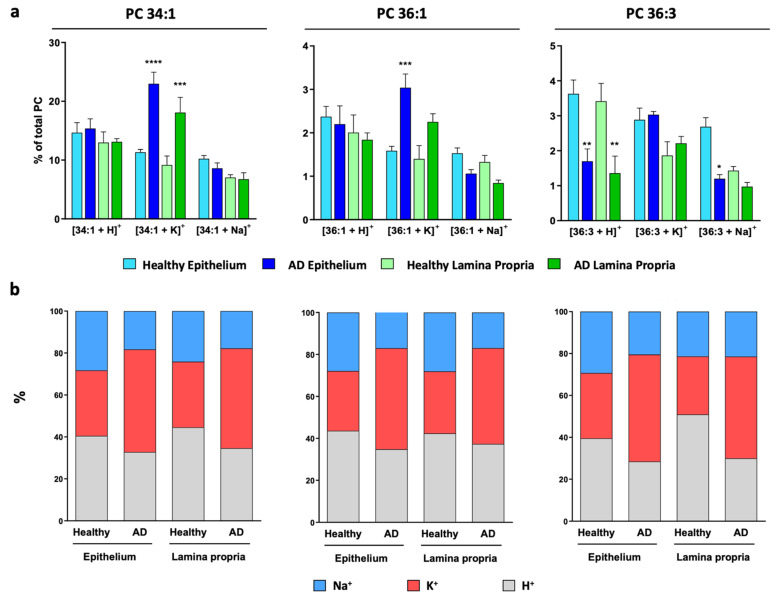
Homogenous impact of tumorigenesis on K^+^ PC adducts. (**a**) Levels of lipid adducts of selected lipid species PC 34:1, PC 36:1, and PC 36:3 in healthy and AD mucosa (the rest of the species may be found in Appendix A). Values are expressed as mean +/− SEM, *n* = 4–5. Adenomatous and healthy tissue were compared using paired or unpaired Student’s *t*-test. * *p* < 0.05; ** *p* < 0.01; *** *p* < 0.001; **** *p* < 0.0001. These species were chosen depending on the impact that malignization had on the total species levels. Thus, PC 34:1, PC 36:1, and PC 36:3 levels increased, remained unaltered, or decreased, respectively, in AD mucosa compared to the healthy epithelium [19]. The rest of the species (PC 34:2, PC 36:2, and PC 36:4) may be found in Appendix A. (**b**) Normalized values showing that despite the impact of tissue malignization on the total levels of a particular lipid species, the level of the potassium adduct was always increased in the adenomatous counterpart.

**Figure 5 cancers-13-01350-f005:**
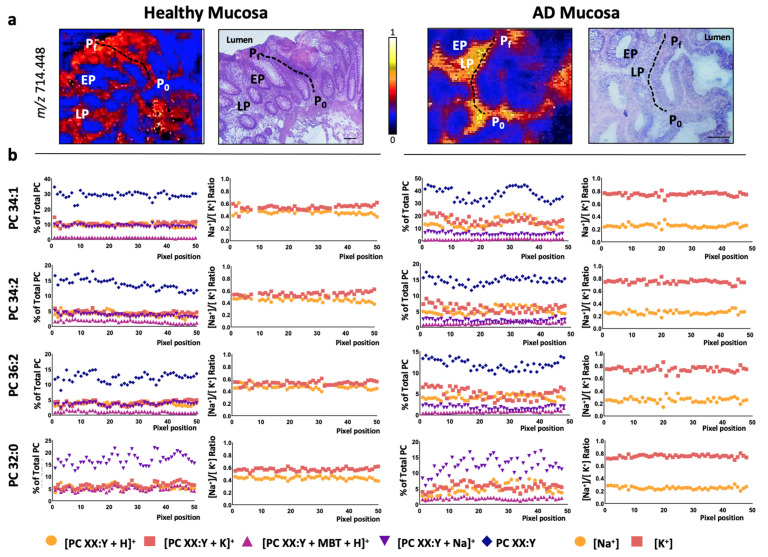
PC adduct distribution in healthy colon and adenomatous polyp lamina propria. (**a**) Selected MALDI-IMS images obtained during the analysis in positive-ion mode (in particular, *m*/*z* = 714.448) showing the paths depicted from the basal (P_0_) to the luminal side (P_f_) of the mucosa to analyze, pixel-by-pixel, the changes in the lipidome along the lamina propria in healthy and AD mucosa. Hematoxylin-eosin images of the consecutive sections are included for comparison. Scale bar = 100 μm. (**b**) Individual distribution of PC adduct along the depicted paths and the [Na^+^]/[K^+^] ratio for selected PC species (PC 34:1, PC 34:2, PC 36:2, and PC 32:0). The rest of the species (PC 36:1, PC 36:3, PC 36:4, and PC 38:4) may be found in Appendix A. EP: epithelium; LP: lamina propria; P_0_: first pixel of the path; P_f_: final pixel of the path.

**Figure 6 cancers-13-01350-f006:**
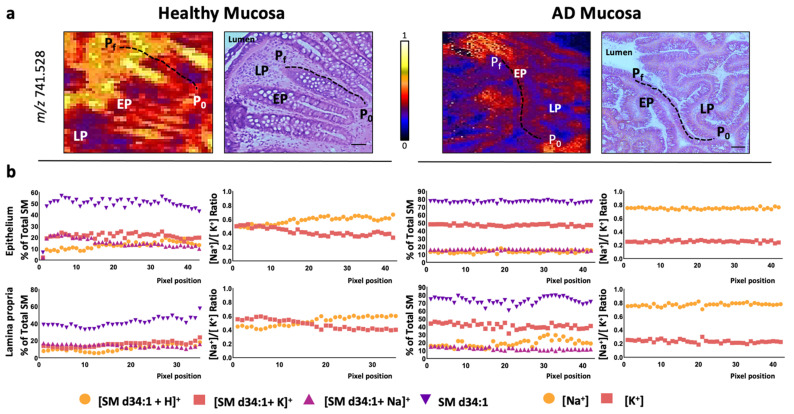
SM-adduct distribution in healthy colon and adenomatous polyp lamina propria. (**a**) Selected MALDI-IMS images obtained during the analysis in positive-ion mode (in particular, *m*/*z* = 714.528) showing the paths depicted from the basal (P_0_) to the luminal side (P_f_) of the mucosa to analyze, pixel-by-pixel, the changes in the lipidome along the lamina propria in healthy and AD mucosa. Hematoxylin-eosin images of the consecutive sections are included for comparison. Scale bar = 100 μm. (**b**) Individual distribution of SMd34:1 adducts along the depicted paths and the [Na^+^]/[K^+^] ratio for selected SM species. EP: epithelium; LP: lamina propria; P_0_: first pixel of the path; P_f_: final pixel of the path.

**Figure 7 cancers-13-01350-f007:**
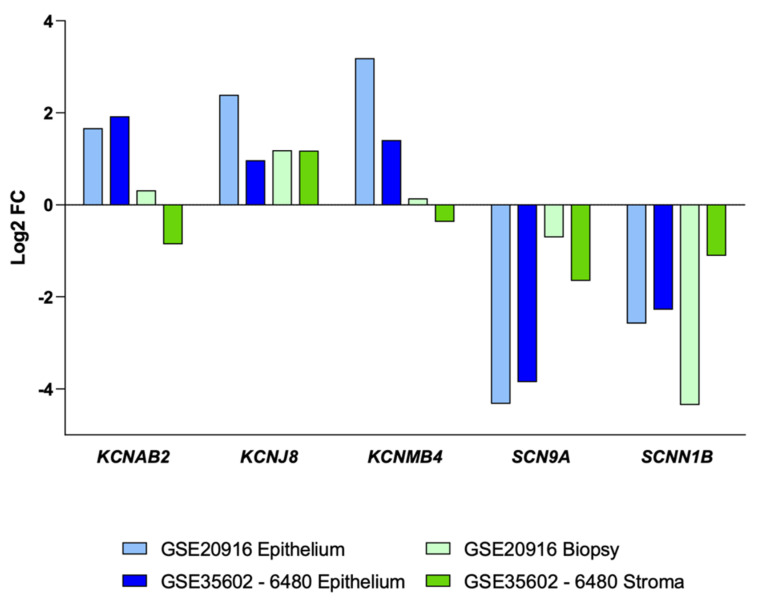
Log_2_ fold change profiles of genes selected as potential targets from different colon CRC tissues compared to their respective healthy control tissues. Data were obtained from the following databases: GSE20916 (*n* = 60 macro-dissected, *n* = 30 micro-dissected samples representing carcinoma and control groups) and GSE35602-6480 (*n* = 34 micro-dissected samples representing CRC and control groups) [32]. Log_2_ fold change ≥1.5 or ≤−1.5, with *p* < 0.05, were considered for CRC gene expression up- or down-regulation. More detailed statistical information on these data can be found in Appendix A.

**Figure 8 cancers-13-01350-f008:**
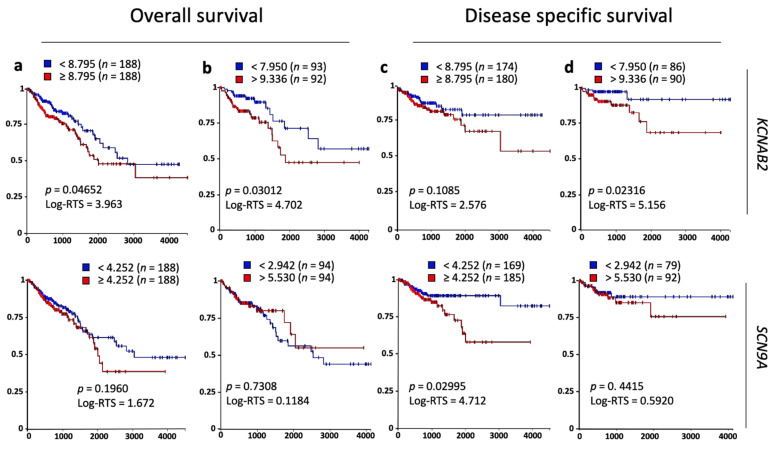
Kaplan–Meier survival analysis in CRC patients based upon KCNAB2 and SCN9A expression in primary colon tumors. Overall survival and disease-specific survival rates (defined in materials and methods) based on UCSC Xena Browser. Log-rank test (test statistics and *p*-value) using two groups of samples (**a**,**c**) or comparing upper and lower quartile (**b**,**d**). RNAseq–Illumina HiSeq gene expression of primary tumor samples (*n* = 380 samples) from TCGA colon and rectal cancer (COADREAD) database. For simplicity, only two genes are included. The rest of the genes may be found in Appendix A.

**Figure 9 cancers-13-01350-f009:**
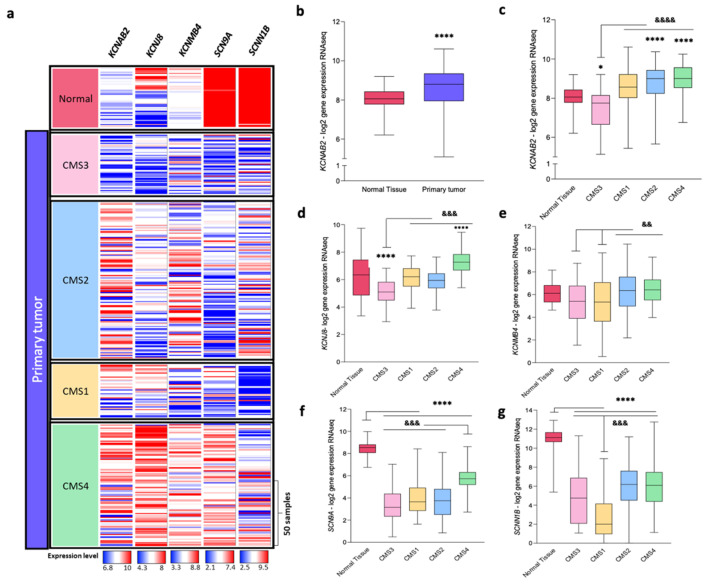
Consensus molecular subtype (CMS) association of gene expression level of potential targets. (**a**) Expression level heat-map of selected genes in primary tumors and solid tissue normal samples, and CMS-sample association [36]. Data were visualized with Xena Browser (USCS) using the TCGA COADREAD database (RNAseq–Illumina HiSeq, *n* = 431). (**b**) *KCNAB2* gene expression in normal and primary tumor tissue. Statistical significance was assessed using an unpaired *t*-test with Welch’s correction, **** *p* < 0.0001. (**c**–**g**) Expression levels of genes coding for *KCNAB2, KCNJ8*, *KCNMB4, SCN9A,* and *SCNN1B,* classified according to the CMS subtype. Statistical significance was assessed using one-way ANOVA. * indicates differences between healthy and the CMS subtypes, while indicates statistical differences among the CMS subtypes to the marked CMS. * *p* < 0.05, ^&&^
*p* < 0.005, ^&&&^
*p* < 0.001, ****^/&&&&^
*p* < 0.0001.

**Figure 10 cancers-13-01350-f010:**
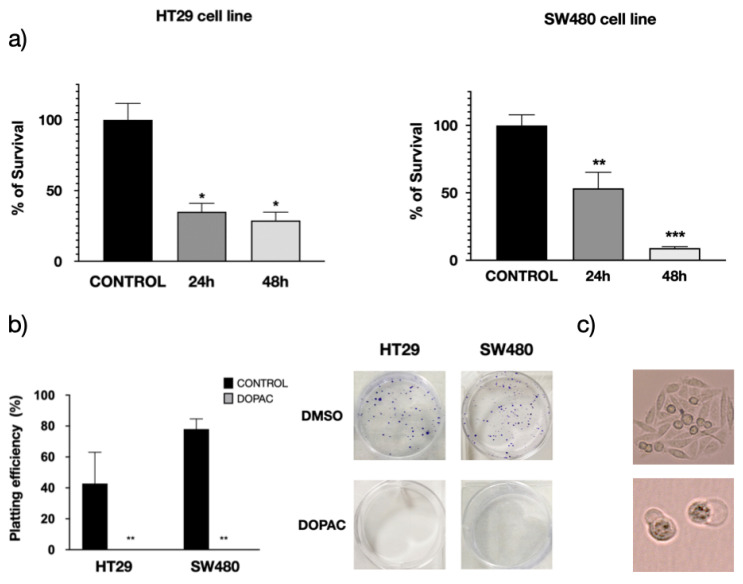
Impact of K^+^ channel pharmacological inhibition CRC cell proliferation. (**a**) Impact of KCNAB2 inhibition measured using the MTT assay. Cells were grown in the presence or absence of DOPAC (500 µM), a KCNAB2 inhibitor [40], for 24 and 48 h. Values represent the mean ± SEM; *n* = 3; statistical significance was assessed using one-way ANOVA comparing treated to control cells. * *p* < 0.05; ** *p* < 0.01; *** *p* < 0.001. (**b**) Clonogenic assay to assess the impact of K^+^-channel pharmacological inhibitors on CRC cell proliferation. Values represent the mean ± SEM; *n* = 3; statistical significance was assessed using unpaired Student’s *t*-test comparing treated to control cells. ** *p* < 0.01. (**c**) Images of the culture dish clearly showing that the DOPAC treatment significantly suppressed clonal expansion in both cell lines. In the last column, insets showing detailed images showing the presence of cells in both control and treated cells. Pictures were taken by Cell Observer, Zeiss (Oberkochen, Germany), magnification ×20.

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
