# Peer review of "A Drastic Shift in Lipid Adducts in Colon Cancer Detected by MALDI-IMS Exposes Alterations in Specific K+ Channels"

_cancers, 2021, doi:10.3390/cancers13061350_

Round 1
Reviewer 1 Report
This manuscript is so much more than just an imaging account of lipids in colon cancer. It addresses the data acquired in colon cancer lipid profiles through the changes that occur to the basic physiology leading to the noted pathology, in order to find reliable biomarkers. They achieved it by correlating the degree of malignancy to shifts in the [Na+]/[K+] ratio, due to altered function of the ion channels accounting for K+ transport.
By investigating the [Na+]/[K+] PC adduct ratio changes pertaining to the colonocyte differentiation status. They then demonstrated a consistent increase in K+ adduct levels, resulting in a shift of the [Na+]/[K+] ratio from 0.5:0.5 to 0.3:0.7. for all PC species. They then made their finding irrefutable as whether or not the changes are specific to PC species only by investigating the changes in sphingomyelin species as SM which also contain quaternary amines and are the second most abundant membrane lipid detected as they ionize easily in positive-ion mode. SM-adducts behaved identically to PC-adducts demonstrating that alterations in [Na+]/[K+] adduct ratio was not lipid class-specific and was a reliable index as to the degree of malignancy of the tumors investigated.
Author Response
We would like to sincerely thank the reviewer for the time taken to review our manuscript and for the positive comments.
Reviewer 2 Report
The manuscript by Garate et al. debuts an elegant elaboration of a simple observation of selective adductization of certain cellular lipids during formation of malignant colon cancer and extends it to the discovery of a potential ion channel candidate for targeted disruption. The manuscript overall is well written, the quality of data is high and convincing, and logical experiments to validate observations have been proposed. The observations are of importance to the readers of Cancers and the study is novel and relevant.
Major Criticisms:
- A major limitation of the current study is that the results of Fig. 9 inhibition are not evaluated in non-malignant H29/SW480 progenitor, healthy cells containing non-aberrant ion channel levels. I suspect the magnitude of inhibition would be less severe. I strongly recommend performing this experiment and including the results. If this is not feasible, as an alternative, please elaborate on the next steps and potential issues with systemic administration of DOPAC and its cytotoxicity (if any?).
- I encourage authors to include loading plots (normal vs malignent) showing the extent of generality of adultization across ALL lipid species observable that can be assigned to further boost confidence in the generality of selective adductization shown using cherry picked lipids. Naturally, one would not expect every single lipid to be affected by the surplus K+ in malignancy but presenting a general observation may help readers appreciate the role of imaging MS that motivated this elegant study.
Minor issues:
- Readers will benefit from justification of mass range 480-1kDa and why fatty acid range was excluded / not deemed informative
- Background info on selectivity (for particular ion channel(s) etc other targets) and cytotoxicity of DOPAC will benefit reader’s appreciation of the potential feasibility of the proposed approach.
- Page 5, section 3.1 : a cartoon representation may be helpful to illustrate important colon structures
- 1: It is unclear what kind of ms images are shown in panel a. Further, histology images lack a one to one correspondence with ms images and key colon cellular structures are not labeled on the H&E. The Fig. 1 legend needs to be revised and the pixel path also shown on the H&E. I suspect H&E is from SAME slide after MALDI (if not, please clarify).
- 1, Fig. 2: There appears to be a shift in path (from edge of the yellow structure, include a heat map in Fig. 1 to middle of the yellow marked structures in Fig. 2. As it is unclear what these mass spec images show and H&E’s are somewhat shifted, it misleads readers to inquire about potential artifact of histologic structure mis-alignment being at play in aberrant K/Na adduct ratios.
- 3: Label basal and laminar sides on H&E and address the issue of shift between MS image and H&E in “AD Mucosa” panels.
- 4 is presented but its results are discussed only in Discussion and after Fig. 5. Add a brief section to “Result”.
- 5. It appears that the left panel is not new data and it reproduces data shown in Fig. 1. If this is the case clearly mention this panel is reproduced and not new data. Same thing appears in a number of SI figures as well that contain graphics from main text figures.
- 6: Text refers to red and purple bars but no such colors are used in Fig. 6. Please revise the text.
- There are two Fig 7b panels and only ONE is discussed in text. Please revise. In the legend, elaborate on the difference between disease specific and overall survival. Which one is used and both values are presented?. It appears that disease specific survival is what most relevant to the work presented.
- 8a results are not presented in “Results”. I recommend add a section and reorganizing such that all figures appear in the same order that they have been discussed during the text. Also, please improve the quality of Fig. 8, It is impossible to distinguish between “*” and “&” and this is important for significance.
- Page 13 first paragraph (above 4. Discussion): There is one long convoluted sentence with its message unclear. Please break down and revise for better clarity.
- Are there any other examples of adductization of analytes being important in MS beyond references 22-24?. There appears to be more (PMID: 26138213) that could potentially be added in support of the observations
Author Response
Response to Reviewer 2 Comments
The manuscript by Garate et al. debuts an elegant elaboration of a simple observation of selective adductization of certain cellular lipids during formation of malignant colon cancer and extends it to the discovery of a potential ion channel candidate for targeted disruption. The manuscript overall is well written, the quality of data is high and convincing, and logical experiments to validate observations have been proposed. The observations are of importance to the readers of Cancers and the study is novel and relevant.
First of all, we would like to sincerely thank the reviewer for the time taken to review our manuscript and for all the constructive comments. Next, we will proceed to address, point by point, all the issues raised by the reviewer.
Point 1: A major limitation of the current study is that the results of Fig. 9 inhibition are not evaluated in non-malignant H29/SW480 progenitor, healthy cells containing non-aberrant ion channel levels. I suspect the magnitude of inhibition would be less severe. I strongly recommend performing this experiment and including the results. If this is not feasible, as an alternative, please elaborate on the next steps and potential issues with systemic administration of DOPAC and its cytotoxicity (if any?).
Response 1: Before answering directly to this important issue, we would like to comment that, in our opinion, the results that support KNCAB2 as a potential pharmacological target to be studied in an oncological context are those showing a consistent increase in the expression of the gene coding for KCNAB2 in primary tumors compared to healthy tissue (Figure 9 in the current reviewed version). We feel is important to emphasize that, in our conclusions, we do not propose DOPAC as a potential antitumor drug. Instead, we used it to provide an in vitro evidence that decreasing the activity of this channel does have an impact on CRC cells. To the best of our knowledge, DOPAC is currently the best inhibitor of this channel according to the study of Alka et al (doi: 10.1016/j.biocel.2014.07.013), which tested the inhibition capability of the compound using purified protein.
On the other hand, in previous studies we used a primary colon epithelial cell line, which are provided as non-tumor cell lines (doi: 10.3390/cancers12051293), to establish their lipidome. In our experience, the characteristics of the commercial non-tumor colon cell lines differ considerably from healthy colonocytes, particularly if they are comparted with colon organoids. Hence, the conclusions using these cell lines should be also carefully considered. Certainly, any KCNAB2 inhibitor that could be designed to be used for human treatment, would need to be tested first for toxicity in non-tumor cells and in animal models.
In this sense, several studies have addressed using different study models the issue of DOPAC toxicity. Interestingly, DOPAC (3,4-dihydroxyphenylacetic acid) is a metabolite of the dopamine, which has been detected in neurons (doi: 10.1080/00365510152567121) and is a regular constituent of rat brain tissue (doi: 10.1016/0014-2999(76)90331-9). DOPAC has been also identified in intestinal microbiota as a fermentation product of the quercetin, which also has potent antioxidant and antiproliferative properties (doi: 10.3390/molecules25184138 and references included). Finally, Zabela et al. studied the pharmacokinetic properties of DOPAC in (doi: 10.1016/j.fitote.2020.104526 ). In this studies DOPAC was given by the intravenous injection in the doses of 1, 2, and 4 mg/kg body weight. The pharmacokinetic studies showed a fast distribution into peripheral tissues and rapid elimination from the body. All these studies seem to indicate that DOPAC had a low toxicity, if any, in non-tumor cells.
Point 2: I encourage authors to include loading plots (normal vs malignant) showing the extent of generality of adductization across ALL lipid species observable that can be assigned to further boost confidence in the generality of selective adductization shown using cherry picked lipids. Naturally, one would not expect every single lipid to be affected by the surplus K+ in malignancy but presenting a general observation may help readers appreciate the role of imaging MS that motivated this elegant study.
Response 2: Regarding this point, we would like to better explain that we have included all the PC species from which we could consistently detect the intensity of both the potassium and sodium adducts. All together 8 PC species fulfilled this criterion: PC 34:1, PC 34:2, PC 36:2, and PC 32:0, which appeared in the main manuscript, and PC 36:1, PC 36:3, PC 36:4, and PC 38:4, which, for the sake of space, were included in the Supplementary Information section (Fig S2, S3, S4 and S5). Although this was indicated in the figure legend, in this reviewed version we have been more specific in the description. According to our results, all the detected PC species experienced a certain increase in K+ adducts (Fig. 4 and Fig S). This result would be consistent with the hypothesis of the study that the misfunctioning of the potassium channels leads to an accumulation of K+ ion, most probably in the extracellular space.
Point 3: Readers will benefit from justification of mass range 480-1kDa and why fatty acid range was excluded / not deemed informative.
Response 3: The main reason is because the fatty acids detected by MALDI-MS could be result of the fragmentation of other lipid species. This is particularly important for our research, as our global aim is to understand the role of membrane lipids in pathology. Hence, the detection of fatty acids of uncertain origin (free fatty acids, glycerolipids, sphingolipids or glycerophospholipids) would hamper considerably the conclusions that we may reach in terms of lipid metabolism. Lipid fragmentation is certainly a major concern for us, and it has been thoroughly studied by some of the authors of this manuscript in:
Influence of Lipid Fragmentation in the Data Analysis of Imaging Mass Spectrometry Experiments. Jone Garate, Sergio Lage, Lucía Martín-Saiz, Arantza Perez-Valle, Begoña Ochoa, M Dolores Boyano, Roberto Fernández, José A Fernández. J Am Soc Mass Spectrom. 2020 Mar 4;31(3):517-526. doi: 10.1021/jasms.9b00090.
In addition, the sensitivity of the ionic trap decreases as the width of the observation window increases. For all these reasons, we focused our attention on the detection of main membrane lipids, being aware that changes in free fatty acid levels do have a role in cell physiology.
Point 4: Background info on selectivity (for particular ion channel(s) etc other targets) and cytotoxicity of DOPAC will benefit reader’s appreciation of the potential feasibility of the proposed approach.
Response 4: This issue has been addressed in Response 1. In addition, we have included part of the information in the manuscript.
Point 5: Page 5, section 3.1: a cartoon representation may be helpful to illustrate important colon structures
Response 5: As suggested, we have included a new figure, Figure 1, summarizing the colon mucosa structures relevant for this study.
Point 6: 1: It is unclear what kind of ms images are shown in panel a. Further, histology images lack a one to one correspondence with ms images and key colon cellular structures are not labeled on the H&E. The Fig. 1 legend needs to be revised and the pixel path also shown on the H&E. I suspect H&E is from SAME slide after MALDI (if not, please clarify).
Response 6: The images originally included were chosen because they were the images where the differences between epithelium and lamina propria were sharper. For this particular purpose, among all the images we obtain in each experiment (one per peak detected), we chose the one showing most clearly the differences between healthy and lamina propria. Doing so, we are certain that all the pixels included in the analysis belongs to the tissue we want to analyze. These differences are less sharp if the content of that particular lipid is similar in a group of cell types, as it happens for example for SMd34:1 (new Figure 6). In this reviewed version, we have included the value of the m/z and the lipid species assigned.
We have also incorporated the improvements suggested by the reviewer. Thus, we have identified the epithelium and the lamina propria in both, the IMS and the HE images; we have also depicted the path in both types of images.
Finally, the reviewer is correct. The HE images correspond to the section consecutive to the one used for MALDI-IMS measurements, as it is mentioned in the figure legends. That is why the structures do not overlap 100%, even if the distance between tissue sections was only 10 µm. Despite the differences, the structures are quite conserved. In this reviewed version we have worked on the HE images adjusting them as much as possible to the IMS image.
Point 7: 1, Fig. 2: There appears to be a shift in path (from edge of the yellow structure, include a heat map in Fig. 1 to middle of the yellow marked structures in Fig. 2. As it is unclear what these mass spec images show and H&E’s are somewhat shifted, it misleads readers to inquire about potential artifact of histologic structure mis-alignment being at play in aberrant K/Na adduct ratios.
Response 7: If by “the shift in the path” the reviewer meant that the paths are different between figure 1(healthy epithelium) and 2 (adenomatous epithelium), that would be correct. In fact, the paths we depict are specific and unique for each of the tissue sections analyzed. Even in the case of the healthy mucosa (figure 1), where the regular structures of the mucosa are maintained, it is not easy to obtain sections showing crypts as the ones showed in the cartoon. To obtain the biopsies the endoscopist pinches off part of the mucosa (a soft tissue) and the tissue is snap frozen into liquid nitrogen. During tissue sectioning with the cryostat, we try to identify the luminal (external) side and then orientate them so that complete crypts could obtained. The situation is more complicated for the adenomatous polyps (figure 2), where the regular structure of the crypt is partially or completely lost. In this case the sections obtained are very different depending on the patient and lesion. Despite of this, we are always able to identify the basal and the luminal side of the mucosa, as well as the pixels belonging to the epithelium and those belonging to the lamina propria, and this is what we use to depict the paths. We followed the same strategy in Bestard-Escalas J. et al 2016 and Lopez, D. et al 2018. We hope that this explanation addresses correctly the concern of the reviewer.
Point 8: 3: Label basal and laminar sides on H&E and address the issue of shift between MS image and H&E in “AD Mucosa” panels.
Response 8: Basal and luminal side of the mucosa have been labelled in the HE images. The issue regarding the apparent shift has been addressed in the previous point.
Point 9: 4 is presented but its results are discussed only in Discussion and after Fig. 5. Add a brief section to “Result”.
Response 9: We have included one paragraph between figures 4 and 5.
Point 10: 5. It appears that the left panel is not new data and it reproduces data shown in Fig. 1. If this is the case clearly mention this panel is reproduced and not new data. Same thing appears in a number of SI figures as well that contain graphics from main text figures.
Response 10: The reasons to include these IMS images are explained in response 6. In the current version of the manuscript, we have replaced them with images corresponding to SMd34:1. However, the bar graphs are showing the changes in SMd34:1 adducts levels for each condition, so they are new data. Nevertheless, to avoid confusion, we have changed that image and include one showing the tissue distribution of SMd34:1 in both healthy and AD mucosa.
Point 11: 6: Text refers to red and purple bars but no such colors are used in Fig. 6. Please revise the text.
Response 11: The text has been corrected as suggested by the reviewer.
Point 12: There are two Fig 7b panels and only ONE is discussed in text. Please revise. In the legend, elaborate on the difference between disease specific and overall survival. Which one is used and both values are presented? It appears that disease specific survival is what most relevant to the work presented.
Response 12: According to the reviewer’s comments, we have made the following changes:
- we have included the definitions of disease specific and overall survival according to NIH in the material and methods section; both parameters were used to draw our conclusions.
- we have labelled the graphs from a to d;
Point 13: 8a results are not presented in “Results”. I recommend add a section and reorganizing such that all figures appear in the same order that they have been discussed during the text. Also, please improve the quality of Fig. 8, It is impossible to distinguish between “*” and “&” and this is important for significance.
Response 13: We have followed the suggestions of the reviewer, reorganizing the paragraph and improving the figure.
Point 14: Page 13 first paragraph (above 4. Discussion): There is one long convoluted sentence with its message unclear. Please break down and revise for better clarity.
Response 14: We have changed the paragraph according to the reviewer’s suggestion.
Point 15: Are there any other examples of adductization of analytes being important in MS beyond references 22-24?. There appears to be more (PMID: 26138213) that could potentially be added in support of the observations
Response 15: The presence of altered levels in potassium adducts of PC species in tumor tissues have been detected previously by other researchers (10.1016/j.neo.2016.07.002; 10.1245/s10434-015-4459-6; 10.1111/cas.12221; 10.1002/pros.23088). However, the changes are usually attributed only to changes in the lipid metabolism, which for sure is involved, but not to an alteration in ion transport. We will include this sentence together with the reference indicated by the reviewer in the discussion.
Reviewer 3 Report
A Drastic Shift in Lipid Adducts in Colon Cancer Detected by MALDI-IMS Exposes Alterations in Specific K+-channels, by Jone Garate et al.
Authors studied the [Na+]/[K+] adduct ratio, as a biomarker for CMC prognosis, by imaging mass spectrometry (MALDI-IMS) techniques. Since alterations in such a ratio is associated with deregulation in K+-channel gene expression, they propose that pharmacologic targets can be identified by using lipid adduct information generated by MALDI-imaging techniques.
The experiments were properly designed and conducted resulting the manuscript an original piece of work; it’s pretty interesting whether from a lipid adduct determined by a sensitive technique, authors are able to associate a real deregulation in K+ homeostasis. The results as well as supplementary figures presentation is outstanding (some missing color is described below). Experimental findings are adequately and justified within result section. Discussion is adequately.
Some questions should be addressed to complete work relevance. My concern is related to the experiments in figure 9. In cell lines, authors studied DOPAC effect, a KCNAB2 inhibitor, on cell proliferation. Regarding this issue, please answer the following:
- Do cell lines express regulatory beta subunit 2 of potassium voltage-gated channel? Have you determined it? Perhaps, PCR assay can answer this question and establish whether KCNAB2 operates deregulating K+ homeostasis in vitro. Have you discarded the toxic effect of DOPAC over its selective inhibitory action on K+ channel? You can show the specificity of DOPAC effect by performing a pharmacological assay by determining cell number as a function of DOPAC concentration.
- MTT always measures mitochondrial functionality, which is not necessarily synonymous of cell proliferation, since many treatments induce mitochondrial biogenesis or are mitochondrial venoms that would kill cells by different mechanisms (e. Sci Rep. 2018; 8: 1531; Adv Clin Exp Med 2008, 17, 5, 525–529). Have you determined cell cycle phases or LDH release or counted cells with exclusion-assays after DOPAC treatments?
- Most important issue:
All these assays show whether DOPAC capability of blocking K+ channels, impairs cell proliferation, but it’s just been reported by others.
Since you state: … Identification of pharmacologic targets using lipid adduct information emphasizes the great potential of IMS lipidomic techniques in the clinical field... you should try to mimic in vivo experiments in culture cells, that is, establish a relationship between [Na+]/[K+] PC ratio and K+ homeostasis deregulation.
The relevance of results in figure 9 should be their association to lipid profile. Thus, have you determined [Na+]/[K+] adduct ratio in culture cell lines? Are you able to associate expression of KCNAB2 with alterations in [Na+]/[K+] PC adduct ratio? Perhaps, you can perform experiments overexpressing KCNAB2 adding different [K+] to media, then, evaluate lipid profile.
Whether or not you perform these experiments (which it would be optimal, giving the information in the main text or as supplementary figure) you have to discuss all the above queries in the manuscript.
Minor concerns.
Introduction, page 3, second line: after coma, we should be deleted.
Results. Page 10, second paragraph and figure 6: bar colors indicated in text are different from those in graph.
Author Response
Response to Reviewer 3 Comments
Authors studied the [Na+]/[K+] adduct ratio, as a biomarker for CMC prognosis, by imaging mass spectrometry (MALDI-IMS) techniques. Since alterations in such a ratio is associated with deregulation in K+-channel gene expression, they propose that pharmacologic targets can be identified by using lipid adduct information generated by MALDI-imaging techniques.
The experiments were properly designed and conducted resulting the manuscript an original piece of work; it’s pretty interesting whether from a lipid adduct determined by a sensitive technique, authors are able to associate a real deregulation in K+ homeostasis. The results as well as supplementary figures presentation is outstanding (some missing color is described below). Experimental findings are adequately and justified within result section. Discussion is adequately. Some questions should be addressed to complete work relevance. My concern is related to the experiments in figure 9. In cell lines, authors studied DOPAC effect, a KCNAB2 inhibitor, on cell proliferation. Regarding this issue, please answer the following:
First of all, we would like to sincerely thank the reviewer for the time taken to review our manuscript and for all the constructive comments. Next, we will proceed to address, point by point, all the issues raised by the reviewer.
Point 1: Do cell lines express regulatory beta subunit 2 of potassium voltage-gated channel? Have you determined it? Perhaps, PCR assay can answer this question and establish whether KCNAB2 operates deregulating K+ homeostasis in vitro. Have you discarded the toxic effect of DOPAC over its selective inhibitory action on K+ channel? You can show the specificity of DOPAC effect by performing a pharmacological assay by determining cell number as a function of DOPAC concentration.
Response 1:
- Yes, these cell lines do overexpress KCNAB2. We apologize because, although we had included the results in Figure S1, they were mentioned only in the Material and Methods section. In this new version, we have mentioned them in the results sections as well.
- Regarding DOPAC toxicity, several studies have addressed using different study models this important aspect. Interestingly, DOPAC (3,4-dihydroxyphenylacetic acid) is a metabolite of the dopamine, which has been detected in neurons (doi: 10.1080/00365510152567121) and is a regular constituent of rat brain tissue (doi: 10.1016/0014-2999(76)90331-9). DOPAC has been also identified in intestinal microbiota as a fermentation product of the quercetin, which also has potent antioxidant and antiproliferative properties (doi: 10.3390/molecules25184138 and references included). Finally, Zabela et al. studied the pharmacokinetic properties of DOPAC in (doi: 10.1016/j.fitote.2020.104526 ). In this studies DOPAC was given by the intravenous injection in the doses of 1, 2, and 4 mg/kg body weight. The pharmacokinetic studies showed a fast distribution into peripheral tissues and rapid elimination from the body. All these studies seem to indicate that DOPAC had a low toxicity, if any, in non-tumor cells.
- Finally, it is worth mentioning that in the clonogenic assay, we were able to detect living cells in the DOPAC- treated wells after more than 10 days of treatment, in fact, the number of cells plated at the beginning of the experiment, rather remained constant all through the experiment. Figure 9c, show a representative image of how cells look like after more than 10 days of experiment.
Point 2: MTT always measures mitochondrial functionality, which is not necessarily synonymous of cell proliferation, since many treatments induce mitochondrial biogenesis or are mitochondrial venoms that would kill cells by different mechanisms (e. Sci Rep. 2018; 8: 1531; Adv Clin Exp Med 2008, 17, 5, 525–529). Have you determined cell cycle phases or LDH release or counted cells with exclusion-assays after DOPAC treatments?
Response 2: We appreciate very much the comment of the reviewer regarding the MTT assay. We have rephrased the paragraph describing the MTT analysis and the results that one may expect.
In this particular study, we did not determine cell cycle phases or LDH release. However, to obtain additional information regarding the impact of DOPAC on CRC cells, we run the clonogenic assay, which assesses the ability of a single cell to grow into a colony. In our opinion, the results reinforced the hypothesis that DOPAC was affecting the capacity of these CRC cell lines to proliferate.
Point 3: Most important issue: All these assays show whether DOPAC capability of blocking K+ channels, impairs cell proliferation, but it’s just been reported by others.
Since you state: … Identification of pharmacologic targets using lipid adduct information emphasizes the great potential of IMS lipidomic techniques in the clinical field... you should try to mimic in vivo experiments in culture cells, that is, establish a relationship between [Na+]/[K+] PC ratio and K+ homeostasis deregulation.
The relevance of results in figure 9 should be their association to lipid profile. Thus, have you determined [Na+]/[K+] adduct ratio in culture cell lines?
Are you able to associate expression of KCNAB2 with alterations in [Na+]/[K+] PC adduct ratio? Perhaps, you can perform experiments overexpressing KCNAB2 adding different [K+] to media, then, evaluate lipid profile.
Whether or not you perform these experiments (which it would be optimal, giving the information in the main text or as supplementary figure) you have to discuss all the above queries in the manuscript.
Response 3: Regarding this issue we have done the following:
- We have decreased the level of enthusiasm regarding the identification of a therapeutic target as one of the conclusions of the study, by adding the word “potential” therapeutic target in the abstract and during the manuscript. However, we still consider the data mining analysis done using large human databases on gene expression in healthy and primary tumors support the candidacy of KCNAB2 as potential therapeutic target worth studying in an oncological context.
- We have also clarified and specified that the aim of the experiments in cells was to provide evidence that the KCNAB2 inhibition affected the growth of CRC cells. We feel is important to emphasize that, in our conclusions, we do not propose DOPAC as a potential antitumor drug.
- Unfortunately, we ruled out the possibility to do study the Na+/K+ ratio PC adducts it in positive-ion mode because the cell culture media contains buffers containing sodium and potassium salts, which would interfere in the adduct levels. In fact, in methodologies as liquid chromatography, ion adducts are formed because of the use of plastic material or buffers. Although we have not analyzed cultured cells directly grown on a plastic dish, we did analyze them as the constitutive part of xenographs. Thus, xenographs of commercial glioma cells were grown in mice, subcutaneously, using commercial cells lines. The MALDI-IMS analysis demonstrated that there was an increase in K+ adducts in the viable areas of the xenografts (Fernández et al 2016 doi: 10.1007/s13361-015-1268-x).
Minor concerns.
Point 4: Introduction, page 3, second line: after coma, we should be deleted.
Response 4: We have corrected the mistake
Point 5: Results. Page 10, second paragraph and figure 6: bar colors indicated in text are different from those in graph.
Response 5: We have corrected the mistake in the reviewed version.
Reviewer 4 Report
This manuscript describes how the [Na+]/[K+] adduct ratio of PC drastically shifts as detected by IMS in cancerous colon mucosa and suggest that KCNAB2 could be an interesting pharmacological target. This work is similar to work done previously with IMS and sodium and potassium ratios on other tissues in other disease. It's application is interesting in the field of colon cancer and the authors have provided some context for why they see this increase in potassium and now KCNAB2 could be an interesting pharmacological target.
Author Response
Response to Reviewer 4 Comments
Point 1: This manuscript describes how the [Na+]/[K+] adduct ratio of PC drastically shifts as detected by IMS in cancerous colon mucosa and suggest that KCNAB2 could be an interesting pharmacological target. This work is similar to work done previously with IMS and sodium and potassium ratios on other tissues in other disease. It's application is interesting in the field of colon cancer and the authors have provided some context for why they see this increase in potassium and now KCNAB2 could be an interesting pharmacological target.
Response 1: We would like to sincerely thank the reviewer for the time taken to review our manuscript and for the positive comments.
Round 2
Reviewer 2 Report
Thank you for your attempts to address my comments.